# Development of a Framework for the Communication System Based on KNX for an Interactive Space for UX Evaluation

**DOI:** 10.3390/s23239570

**Published:** 2023-12-02

**Authors:** Ariel A. Lopez-Aguilar, M. Rogelio Bustamante-Bello, Sergio A. Navarro-Tuch, Arturo Molina

**Affiliations:** School of Engineering and Sciences, Tecnologico de Monterrey, Mexico City 14380, Mexico; snavarro.tuch@tec.mx (S.A.N.-T.); armolina@tec.mx (A.M.)

**Keywords:** home automation, User Experience (UX), sensory environment, KNX, emotion recognition

## Abstract

Domotics (Home Automation) aims to improve the quality of life of people by integrating intelligent systems within inhabitable spaces. While traditionally associated with smart home systems, these technologies have potential for User Experience (UX) research. By emulating environments to test products and services, and integrating non-invasive user monitoring tools for emotion recognition, an objective UX evaluation can be performed. To achieve this objective, a testing booth was built and instrumented with devices based on KNX, an international standard for home automation, to conduct experiments and ensure replicability. A framework was designed based on Python to synchronize KNX systems with emotion recognition tools; the synchronization of these data allows finding patterns during the interaction process. To evaluate this framework, an experiment was conducted in a simulated laundry room within the testing booth to analyze the emotional responses of participants while interacting with prototypes of new detergent bottles. Emotional responses were contrasted with traditional questionnaires to determine the viability of using non-invasive methods. Using emulated environments alongside non-invasive monitoring tools allowed an immersive experience for participants. These results indicated that the testing booth can be implemented for a robust UX evaluation methodology.

## 1. Introduction

Domotics, also referred as Home Automation, is the integration of controlling and monitoring systems in the home area in a unified system allowing the transition of conventional homes into smart homes [1]. The goal of smart homes is to monitor inhabitants’ activities through the interaction with different devices in the home area to control and adapt the environment in order to provide a better experience for its inhabitants [2].

Domotics has a variety of devices that can be programmed according to the application context, demonstrating the potential to be adapted to different scenarios such as education [3], healthcare [4], assisted living [5], etc. Domotics is also used for relaxation therapies to reduce stress by generating scenarios with smart lighting systems, environmental sound players, temperature variation, and scent diffusers [6,7,8,9]. These applications inspired us to create the Emotional Domotics (ED) research line. The motivation behind ED research is to improve the quality of life by reducing the stress levels of the inhabitants with domotic systems. This goal can be achieved by creating an inhabitable space that monitors and analyzes inhabitants’ emotional responses leading to the selection of the best environmental condition in which a sense of comfort and well-being is transmitted to the inhabitants, reducing stress levels as a result [10]. Considering this, a testing booth was built to conduct experiments centered around the impact of the environmental variables in the emotional behavior of the inhabitants. The testing booth was instrumented with KNX devices to facilitate the replication of experiments; KNX is an instrumented standardized open-source communication protocol for building automation and domotics. Meanwhile, the emotion recognition task is carried out using non-invasive methods such as wearables and computer vision to acquire biometric data from the inhabitants [11]. There are similar works that implement smart home systems to recreate scenarios and use physiological sensors such as wearables to monitor stress levels [12,13,14]. In these works, predefined scenarios are generated for relaxation therapies, while stress levels are calculated using signals acquired from electrodermal activity and heart rate. Based on the implementation of these solutions, inhabitants benefit from selecting the scenarios that are more suitable for them.

Meanwhile, as this research progressed, we established the possibility of using the testing booth in UX research due to the generation of environments according to product design and testing in conjunction with the emotion recognition systems to analyze users’ behavior during the interaction [15]. The application of the methods and tools used in the testing booth can also improve the efficiency of individuals in workstations [16] and student engagement in learning environments [17]. However, the KNX communication protocol was not designed to integrate sensors to monitor users’ emotional response, but it can integrate biometric sensors for security purposes such as fingerprint scanners. This led us to develop a framework for the communication system of the testing booth to integrate KNX-based systems alongside with emotion recognition systems. The framework was designed based on the Python programming language, taking advantage of the open-source condition of the KNX communication protocol in order to integrate and synchronize data collected from the environmental sensors with user data collected from biometric sensors during the interaction process. Using Python also allows the possibility to integrate Artificial Intelligence (AI) and Machine Learning (ML) tools for a more advanced emotion recognition system and facilitate the control when recreating scenarios for product testing. A diagram of the general solution based on the requirements of the framework of the communication system of the testing booth is presented in Figure 1. The diagram consists of an interactive space, the testing booth, that contains KNX actuators and sensors. Within the interactive space, different activities are performed related to the interaction between users and products. Biometric data for emotion recognition such as facial expressions are acquired with sensors that do not belong to KNX. The acquired data from the environmental sensors and biometric sensors are synchronized and processed for their interpretation to generate a UX report.

To test this framework, an experiment was proposed in which a new design of detergent bottle was evaluated using the emotional response of the participants. The testing booth was used to recreate the environment of a laundry room (relative humidity, light hue and intensity). The recreation of environments in conjunction with non-invasive monitoring sensors allowed participants to have a more immersive experience. Furthermore, using emotional responses allows researchers to acquire a more genuine response from the participants while they are interacting when evaluating product usability criteria. This is important since UX researchers can make a deeper analysis of the elements that could impact on the acceptance of a new product and reduce any possible bias generated by self-reported metrics such as questionnaires and surveys [18]. These findings can be implemented to design a methodology for an objective UX evaluation based on emotions.

This paper is structured as follows: Section 2 provides the background to understand this research line. Section 3 describes the testing booth as well as the instruments it has. In Section 4, the proposed framework for the communication system of the testing booth is presented. Section 5 covers the application of the proposed framework in an experiment related to the analysis of UX for the evaluation of prototypes of detergent bottles in a simulated scenario. Finally, a discussion about the findings and the implications of this work is provided in Section 6.

## 2. Background

In this section, a brief description about the topics covered in this research is provided.

### 2.1. Domotics

Domotics focuses on transforming conventional homes into smart homes. This means that smart technologies are used to control indoor environments by installing intelligent lighting systems, entertainment systems, temperature controllers, and other applications to improve the comfort and safety for any user [19,20]. All mechanical and digital devices are interconnected to a network allowing the communication with each other and with the final user to create an interactive space [21]. As previously mentioned, the main objective of domotics is to improve the quality of life of inhabitants by automatizing most home tasks. This can be achieved by providing a proactive environment that is aware of their inhabitants’ personal and emotional needs based on their location in the smart home, and it can have several solutions for those specific needs [22]. Every smart home system must have the following features [20]:Automation: the ability to accommodate automatic devices or perform automatic functions.Multi-functionality: the ability to perform several duties to generate several outcomes.Adaptability: the ability to adjust to inhabitants’ needs.Interactivity: the ability to interact with or allow interaction among inhabitants.Efficiency: the ability to perform functions in a time-saving, cost-saving, and convenient manner.

As domotics evolves, new tendencies and technologies have emerged. Among those tendencies, there is the implementation of the Internet of Things (IoT) to manage all the different devices in a smart home, facilitating the monitoring of individuals and boosting independent living through sensors and actuators connected to internet networks to manipulate the environment [23]. Meanwhile, IoT systems allow the processing and storage of a great amount of data; this means that the domotic system can take advantage of the collected data, detect patterns on inhabitants’ behavior by implementing ML models, and adjust the environment to routines according to those routines, resulting in the energy efficiency [24]. As all domotic devices are connected by an internet network, inhabitants can monitor and control their smart homes remotely. However, these developments have also led to increased investments in cybersecurity solutions to prevent intruders from having access to the collected data and controlling the smart home [25].

### 2.2. KNX Technology

Every domotic installation must have a proper communication protocol to exchange data between all devices in the home area to control and monitor all the smart home devices. Many communication protocols for domotics have been developed over the years like BACnet, C-Bus, CC-Link, and KNX [26]. Among these communication protocols, the standard KNX is one of the most popular for smart homes and building automation because it provides a standardized communication protocol that allows the exchange of data between KNX domotic devices. The KNX protocol offers backward compatibility, allowing an easy installation and scalability as well as minimizing upgrade costs. Many manufacturers can develop their own KNX devices that must use the same communication protocol; this means that all KNX devices can exchange data regardless of whether they were made by different manufacturers. Additionally, the KNX-based systems offer a variety of solutions for smart lighting, heating systems, energy efficiency, and security systems. Due to these features, KNX has more than a 70% share of the automation market in Europe [27].

The KNX protocol was developed by the KNX Association. This association was founded in 1999 by the European Installation BUS Association, European Home System Association, and BatiBUS Club International. The KNX protocol was accepted as an international standard for home automation in 2006 as ISO/IEC 14543 [28] as well as CENELEC EN50090 [29] and CEN EN 13321-1 [30] (Europe), ANSI/ASHRAE 135 [31] (USA), and GB/T 20954 [32] (China) [33].

The minimum KNX working system must include at least a KNX Power Supply of 30 V DC, an actuator, and a sensor. Data exchange between devices is accomplished by different transmission media such as KNX TP (Twisted Pair) or KNXnet/IP (Ethernet). Any KNX installation can be programmed by using the Engineering Tool Software (ETS) Version 5 [34]. In a KNX installation, devices are identified by a physical address and a group address. The physical address is used to initialize and program the device. Meanwhile, the group address is used for the communication and interaction between KNX devices. This allows an easy and fast exchange of data between KNX devices [35].

Being an open-source communication protocol, KNX opens the possibility for anyone to develop their own solutions for home automation or building automation. This includes the possible adaptations of the communication protocol to other platforms such as MATLAB 2017b (or superior) or Python 3.6 (or superior) for different applications [36,37]. Finally, KNX allows all instruments to be under the control of the same communication protocol; this simplifies the process of adding other instruments that are based on KNX regardless of whether they are from different manufacturers. Due to these features, KNX-based systems were selected to instrument the testing booth. The specifications of the devices used in this research are provided in Section 3.

### 2.3. Emotion Recognition

Emotions are complex mental reactions that can be conscious or unconscious responses to events, objects, and situations. Emotions combine feelings, thoughts, and behaviors that can manifest through various channels, including human speech, gestures, facial expressions, and physiological signals [38]. Emotions can impact on both the physiological and psychological states of individuals [39]. There are different technologies to identify and measure the intensity of emotions. These methods can be divided into contact and non-contact methods. Contact methods require specialized devices to acquire physiological data such as EEG, EMG, GSR, or ECG. Meanwhile, non-contact methods use tools for the processing of videos to analyze facial expressions and body posture as well as audio recordings for voice analysis. In the context of this research, the computer vision algorithms are used for the analysis of facial expressions to identify emotions.

One of the most recognized researchers in facial expression analysis is Dr. Paul Ekman. Ekman and his colleagues suggested that emotions can be consciously or unconsciously expressed through facial expressions. A set of basic emotions can be identified by analyzing facial expressions; these emotions are joy, anger, contempt, disgust, fear, sadness, and surprise [40]. These expressions are innate and consistent across individuals regardless of factors such as age, culture, or ethnic origin [41]. This means that there are similar patterns in facial expressions when an individual is experiencing a specific emotion.

Another key factor is the advances in ML and deep learning models that led to the creation of solutions around computer vision algorithms like emotion recognition using facial expressions analysis for different purposes such as marketing, psychology, education, among others [42]. The methodology to analyze facial expressions for emotion recognition generally follows the steps presented in Figure 2.

The first step consists of acquiring an image that will be analyzed. This image can be obtained from a photo or a video; this image is called the input image. In the pre-processing step, the input image is analyzed to obtain a Region of Interest (ROI). ROIs are used to simplify the processing of the image by giving the algorithm relevant data: in this case, the face of an individual. ROIs are used to crop the areas that contain all possible faces within the input image. Most facial expressions recognition systems are based on the Viola–Jones algorithm [43] or Dlib algorithm [44]. Several filters are applied in the input image based on the architecture of the model; these filters include resizing and changes in color channels.

During the feature extraction phase, relevant data are obtained from the pre-processed image. One of the most popular techniques to obtain facial features is Action Units (AUs). AUs are individual facial muscle movements involved when a subject is expressing an emotion. AUs are based on the theory behind of the Facial Action Coding System proposed by Dr. Paul Ekman [45].

In the classification/regression step, the processed image is labeled with the emotion that the individual is experiencing. The most popular algorithms to classify emotions based on facial expression analysis are Convolutional Neural Networks (CNNs) and Support Vector Machines (SVMs) [46].

### 2.4. User Experience

As previously mentioned, smart homes are adjusted to the users’ needs and requirements in order to provide an active scenario that generates a sense of well-being and improves the quality of life [47]. Therefore, it is important to know user perception when interacting with these systems. Creating new metrics to quantify user acceptance and satisfaction has gained major relevance recently. These new metrics are related to the concept of UX [48].

Although UX is a relevant concept in research, there is not a clear definition about UX and how it should be measured. This problem emerges due to the unique applications given by different disciplines and areas of knowledge [49]. From this discussion, to create a unique concept of ISO 9241-210 [50], ISO 9241-210 standardized the concept of UX as “A person’s perceptions and responses that result from the use or anticipated use of a product, system or service”. This definition is used by new researchers to have a better understanding about UX and how it could be applied to the area of knowledge they are working on [51].

Questionnaires are the most used methods to evaluate UX because it is possible to obtain quantitative data about the interaction with the product or service. However, the obtained data can be biased or impacted due to external influences such as the participant’s attitude or their current emotional state, among others [52]. Therefore, many UX researchers have introduced the use of psycho-physiological methods such as electroencephalography (EEG), electromyography (EMG), or Galvanic Skin Response (GSR) in order to obtain a relative objective UX evaluation [18].

### 2.5. Previous Work in Emotional Domotics

In previous works [10], the impact of environmental variables on a user’s emotional state was analyzed. The first experiments were conducted in a Gilbreth and Taylor testing booth in which participants were asked to assemble a LEGO^®^ vehicle under different environmental conditions; these environmental conditions include light intensity, temperature, and humidity. Each experimental test was designed to last 5 min. The experiments were conducted in Mexico; therefore, all environmental conditions were adjusted according to the Official Mexican Norms: NOM-015-STPS-2001 [53] and NOM-025-STPS-2008 [54], which are related to temperature conditions and lighting conditions, respectively.

iMotions™ software version 8.1 was used to analyze participants’ emotional responses during the experiments [55]. iMotions™ software implements the FACET™ module for the analysis of facial expressions with computer vision for emotion recognition; although more recent versions of iMotions™ implement Affectiva™ model for this task, since FACET™ was acquired by Apple© [56]. The results of these experiments led to the design and construction of a testing booth according to the needs of ED research.

After the testing booth was built, different experiments were conducted. In this case, the experiments were passive; this means that participants received the stimuli instead of performing an activity. The stimuli were obtained from the Interactive Affective Picture System (IAPS) [57]. The experiments were conducted using sensors compatible with iMotions™ software: the FACET™ module and the Shimmer™ wristband to collect GSR and photoplethysmogram (PPG) data. Temperature, light hue and intensity were modified for each experiment in the ranges allowed by the Official Mexican Norms. The results of these experiments were used to generate equations that correlate the emotional state of the user with the environmental variables. These equations are planned to be used to create a control loop which modifies the inhabitable space variables based on the user’s emotional needs [11].

## 3. Methodology

In this section, a deep description about the testing booth is provided. This description includes the KNX devices and how they are installed as well as the process of the interpretation of the KNX protocol for the communication framework.

### 3.1. The Testing Booth

Based on the conclusions from previous experiments [11], it was stated that a testing booth was needed to conduct different experiments to analyze in depth the impact of the environmental variables on users’ emotions. In this case, the environmental variables we are working with are temperature, light intensity and hue, relative humidity, and CO_2_ levels. The testing booth was designed and built with the help of students studying toward a Bachelor degree in Industrial Design as an academic challenge. Aspects such as good light reflection from the inside, low heat transfer, and isolation from any visual distraction from the outside were considered during the design process. The testing booth has the necessary elements that can be used by the participants to perform different tasks. The testing booth has two Microsoft LifeCam HD-3000 webcams: one is used to record the facial expressions in order to analyze them and identify emotions, and the another is used to record the interaction to identify the tasks performed. The testing booth is presented in Figure 3.

The testing booth has a square base of 1.65 m and a height of 2.4 m. Its structure is made of steel, and it is covered with medium-density fiberboard (MFD) panels except for the roof that is composed of panels of polycarbonate.

As previously mentioned, the testing booth was instrumented with different KNX devices:CO_2_ Multisensor CD 2178: Sensor to measure different variables within the testing booth. These variables are CO_2_ levels (parts per million—ppm), relative humidity (%), and temperature (degrees Celsius).Universal presence detector Jung 3361WW KNX: Sensor with a 360-degree detection angle divided into three 120-degree zones to measure light intensity (lux) inside the testing booth.3902 REGHE—KNX 2 channel Universal Dimmer Actuator: This actuator allows changing the light intensity using dimming. The lamps that can be manipulated are incandescent lamps, 230 V halogen lamps, inductive transformers, inductive transformers with low-voltage LED, and dimmable compact fluorescent lamps.BX-DM01—Blumotix KNX: This actuator works as a dimmer for an RGB LED strip. It is a four-channel dimmer actuator that can be configured to work with LED strips of 12 to 24 V; each channel has an output of 4 A.ZN1CL-IRSC—Zenio KNX: This actuator is an infrared control module that allows us to control the air conditioner remotely by means of a series of previously programmed commands without the need to use a remote control. An important point is that the transmitter must be in the line of sight of the infrared receiver of the air conditioner. It has a voltage range of 21 to 31 VDC with a maximum consumption of 10 mA.320 mA Power Supply—JUNG KNX: It supplies 320 mA and controls the system power for the KNX installation. The devices can be connected to that through the BUS line.Communication Module IP: The IP communication module allows access to the system via IP from any PC loaded with the ETS3 or higher or with a visualization software. It works in “Tunnelling” mode and offers up to 4 simultaneous KNXnet/IP connections.

The installation can be controlled using KNXnet/IP protocol using any PC with the ETS5 software [58]. KNX has its own special bus cable to communicate with every KNX device in the building. The bus cable can go alongside the domestic tension cable in a KNX installation as long as both cables are covered with their respective isolation material. The wiring diagram of the testing booth is shown in Figure 4. The wiring diagram consists of the KNX instruments mentioned above and a PC to control the environmental variables within the testing booth. To connect the PC with the instruments of KNX, a router is needed to send control commands to the KNX controller IP interface. Finally, the two webcams are connected to the PC to record the facial expressions and the interaction between the user and product. A monitor is also installed within the testing booth to present stimuli if they are needed.

More KNX devices could be added, but they need to be configured with the ETS5 software. Adding new KNX devices does not affect the settings of previously installed devices.

### 3.2. Interpretation of the KNX Communication Protocol

The communication of the testing booth is based on the KNXnet/IP protocol. The KNXnet/IP protocol is a variant of the KNX protocol that allows data exchange through an IP network using Ethernet as its physical layer. The User Datagram Protocol is used as the transport layer, while the KNXnet/IP serves as the application layer [36].

The KNXnet/IP protocol uses a telegram system to exchange data between all devices in a KNX installation. Telegrams are data packets containing commands with the necessary data to carry out several instructions in a KNX installation [34]. KNX documentation contains information about the content of those telegrams; however, it does not provide a depth description about how data are handled. Wireshark™ was used to obtain the content of KNX telegrams and analyze every line. The methodology to acquire the telegrams in order to examine them consists of five steps:The monitoring is initialized to capture all data packets that belong to the KNXnet/IP protocol; other data packets are discarded.The connection is established with the KNX system to generate all the telegrams related to the connection request from the computer to the KNX IP BUS.Commands are sent to all sensors and actuators installed in the testing booth to register the generated telegrams.A command is sent to disconnect the computer from the KNX IP BUS and identify the telegram corresponding to this command.Monitoring is stopped since all necessary telegrams were captured.

The content of each KNX telegram was analyzed with Wireshark™. This analysis has the purpose to have a better understanding of about how the data are encapsulated in a KNX telegram in order to replicate its behavior.

After analyzing each recorded telegram, it was discovered that telegrams are coded in hexadecimals, and they have the following elements:Header length: It indicates the start of the telegram, and it never varies.Protocol version: It indicates the version of the KNXnet/IP protocol that is being applied.KNXnet/IP service type identifier: It indicates the type of action that will be performed in the KNX installation.Total length: It indicates the number of bytes that the telegram contains.KNXnet/IP body: It contains all the necessary commands to do the requested action.

Once the process of KNXnet/IP protocol was processed to generate the corresponding commands, a programming language was selected to replicate the KNXnet/IP protocol to control the testing booth. In this case, Python was chosen due to the facilities it offers such as tools for computer vision, AI, and interface design, and it can work with asynchronous systems.

In the search for tools and libraries for asynchronous systems, the XKNX library was found; XKNX is an open source library developed by third parties. However, this library is not officially licensed by the KNX Association. This led to conducting different tests in order to verify the communication between KNX devices installed in the testing booth by implementing this library.

To establish a successful connection using this library, the parameters must be set according to the group addresses of the different KNX devices inside the testing booth. The first tests consisted of turning on a white light and modulating its brightness. Wireshark™ software 2.6 was used to monitor this process. How data were handled is shown in the handshake diagram in Figure 5.

From these tests, it was proved that the XKNX library is compatible with all KNX devices installed in the testing booth. With Python, a graphical user interface (GUI) was proposed to control and monitor the environmental variables within the testing booth using the XKNX library.

The GUI has space to show the image captured by the webcam. The image captured by the webcam can be processed by emotion recognition models based on facial expressions analysis. The displayed image has a resolution of 920 × 750 p, and it has a 15 mS refresh rate. Meanwhile, limitations of the actuators were also considered during the programing of the GUI. For example, a range of temperatures was set for the air conditioning in which users can only select temperatures between 20 and 28 °C. All data collected by the sensors, as well as the changes in the actuators are taken every 3 s and saved into a CSV file. This time was set due to there not being any abrupt changes in the environmental variables during a short period of time. Meanwhile, the analysis of facial expressions should be made at a frame rate of 24 FPS. The prototype of the GUI is presented in Figure 6.

## 4. Proposed Framework for the Testing Booth Communication System

With the acquired knowledge during this research and delving into the opportunities of the KNX communication protocol, a framework was proposed for the communication system of the testing booth to integrate the tools for emotion recognition with the devices based on the KNX protocol. The proposed framework for the communication system of the testing booth is presented in Figure 7; this framework is divided into seven main sections.

### 4.1. Interactive Space

The focus of this section is the physical space in which the experiments are conducted. In this case, it includes the testing booth with all its elements: design, furniture, equipment, and electrical systems.

In this interactive space, participants interact with different elements. Experiments will be designed in order to make the participants interact most of the time with the mouse and keyboard, but it is not limited and it is possible to interact with other kinds of objects, as will be presented in Section 5. Passive audiovisual stimuli in the screen are also considered. From these interactions, the emotional response of the participants is acquired by non-invasive means as recordings of the facial expressions. However, other devices such as wearables to acquire physiological signals such as GSR or heart rate are considered for experiments that require them. These signals will be used as complements for the analysis of the emotional response.

### 4.2. Actuators

Actuators are used to control the environment in the interactive space to generate a scenario according to the experiment. There is a white light system and an RGB LED strip to adapt the intensity and the hue of the light. The testing booth also has an air condition system to modify the temperature based on the requirements of the experiment. The air conditioning system has different modes like automatic, cold, dry, and CO${}_{2}$ mode; Official Mexican Norms are considered for the selection of the environmental variables.

### 4.3. Sensors

This section can be divided into environmental sensors and biometric sensors. The environmental sensors are installed in the testing booth to acquire signals such as CO${}_{2}$ levels, relative humidity, temperature, and light intensity.

For the biometric sensors, a 720 p resolution webcam is installed to capture each participant’s facial expressions. The face must be captured frontally as much as possible; therefore, the webcam is placed above the screen in which the audiovisual stimuli is shown. Other sensors such as wearables are included to acquire complementary physiological signals. The selected wearable must allow participants to interact with the elements within the testing booth freely.

### 4.4. Domotic Interface

The domotic interface is the key point to establishing communication among all the devices in the testing booth. This communication uses the KNX IP BUS to send KNXnet/IP telegrams to all KNX devices. The telegrams have the commands to make KNX devices execute different orders as well as to request data from the sensors and state of actuators. The acquired data are sent via a KNX IP BUS to the computer that controls the testing booth.

### 4.5. Data

In this section, the obtained data from the sensors are collected. The variables from the sensors are temperature in degree Celsius, CO${}_{2}$ levels in particles per million (ppm), relative humidity in percentage, and light intensity in lux. Meanwhile, the data acquired by biometric sensors include the GSR and heart rate as well as the recording from the webcam with the participant’s facial expressions.

Additionally, there is a database with the audiovisual stimuli that will be presented to the participants depending on the experiments’ requirement. However, acoustic stimuli will not be used at this stage of the research due to the emotional complexity related to music.

### 4.6. Processing

In this section, participant’s facial expressions are analyzed as well as the acquired physiological data collected by the wearables to complement the processed data from facial expressions. Although there are instruments to recognize emotions with a better accuracy such as EEG, those instruments tend to generate a bias on participants’ behavior, since they feel a constant sense of observation. The process of emotion recognition includes ML algorithms, primarily CNNs, to analyze facial expressions and detect which emotion the participant is experiencing in that moment. The collected data related to emotions, stimuli, and interaction are synchronized frame by frame with the data from the environmental sensors. The synchronized data are analyzed with AI tools to identify patterns in participants’ behavior during the interaction and interpret those patterns to understand the impact of the interaction variables and the environmental variables on participants’ emotional response. A key aspect considered during the design process of this framework was the privacy of the participants since their biometric data are collected; therefore, an informed consent document must be redacted. This document must inform participants how their data will be processed, and it will not be spread to other entities that may misuse the collected data. This extra step guarantees privacy to all participants involved in future experiments.

### 4.7. Application

Once the patterns are interpreted, a UX report is generated based on participants’ emotional behavior during the experiments. The environmental variables are selected for the next iteration of the experiment; therefore, KNXnet/IP telegrams are generated to be sent to the KNX devices in the testing booth to modify the environmental variables to create a scenario for the experiment. Also, the system is programmed to acquire signals from the KNX sensors during a period depending on the requirements of the experiment. In case any device does not receive an instruction, the telegram must be sent again. The KNX protocol has an “Acknowledge” telegram system to inform that a telegram was received successfully by the target device, and it is carrying out the requested task. Finally, the stimuli that will be presented to the user in the scree is selected.

## 5. Application of Proposed Framework for UX Analysis

The testing booth and its framework have the potential to be implemented in applications related to UX research. To evaluate the framework presented in Section 4, an experiment was conducted in order to evaluate the design of detergent bottle prototypes based on the emotional response of participants. Seven detergent bottles were designed and divided into three groups: Group A included 600 mL three detergent bottles, Group B featured 1 gallon detergent bottles, and Group C contained 5 L detergent bottles. The design of the bottles is presented in Figure 8. In this experiment, the testing booth was used to replicate a scenario corresponding to a laundry room in which participants interacted with the prototypes of detergent bottles, while data related to the emotional response of the participants were collected to be analyzed in order to determine the acceptance rate of the new designs of the detergent bottles.

This experiment was conducted in collaboration with students pursuing a Bachelor’s of Industrial Design who designed and created the prototypes of the detergent bottles. They even recreated the weight of the detergent bottles to create the most genuine interaction as possible. Other students that collaborated in this research were students studying toward a Bachelor of Industrial Engineering that helped with the logistics of the experiment. The impact of the involvement of these students and how this experiment contributed to their academic formation is explained in [59].

### 5.1. Development of the Experimental Protocol

The main goal of this experiment was to determine the level of acceptance of the new design of detergent bottles based on the emotional response of participants. To achieve this goal, participants’ facial expressions were analyzed as well as signals such as GSR and PPG (heart rate).

Meanwhile, the interaction with the bottles was also recorded to synchronize them with participants’ emotional responses and identify the key moments in which participants have a more intense emotional reaction. The collected data are sensitive; therefore, an informed consent document was redacted which ensured the anonymity of the participants, and the data related to their facial expressions will be used exclusively for the emotion recognition process, and they will not be used for other purposes.

To collect the data, two cameras were used: one camera to record the facial expressions of the participants and another camera to record the interaction with the detergent bottles. The cameras used for this experiment were two Microsoft LifeCam HD-3000. Facial expressions were analyzed with the FACET model which has an accuracy of 97% when predicting emotions [60]. The data related to GSR and PPG were obtained using Shimmer™ wristbands. Facial expressions and physiological signals were processed with the iMotions™ software.

For this experiment, thirty participants (27 female and 3 male, in an age range between 18 and 43 years) were gathered. The participants were distributed according to the capacity of the detergent bottles from Figure 8. Table 1 presents how the participants were distributed between the bottles; none of the participants repeated the experiment with another group of bottles.

The characteristics of the detergent bottles to be evaluated during the interaction are the following:First impression;Dosage measurement;Bottle’s grip;Spill prevention;Residue prevention.

Based on this characteristic, instructions about the interaction were recorded with a neutral voice. Each instruction was designed in order to make the execution time about two minutes. The estimated time of each experiment per participant was 35 min. In this time, we considered the indications given to the participants about how the experiment will be conducted, the reading and signing of the informed consent document, the instrumentation of the participants, the interaction with the detergent bottles, and an application of a questionnaire about the new designs of the detergent bottles. The results of the questionnaire were contrasted with the emotional response of the participants.

Once experiments were over, the acquired data were interpreted to determine which emotions have a major intensity during the interaction with the detergent bottles; the interpretations were reported as the results of the UX analysis.

### 5.2. Setup of the Testing Booth

As mentioned above, the testing booth was adapted to emulate a laundry room; this means that the environmental conditions such as temperature, illumination, and relative humidity had to be adjusted. The temperature inside the testing booth was between 24 and 26 °C; this margin was acceptable according to the Official Mexican Norm [53]. Relative humidity was adjusted to 50%. Finally, light intensity was adjusted to 93 lux, while light tonality was orange. These parameters were selected according to the typical laundry room in a Mexican context.

### 5.3. Environment–User–Product Interaction Process

Prior to starting each experiment, instructions about the experiment were provided to the participants; this was followed by the reading and signing of the informed consent document. Once the document was signed, participants were fitted with the Shimmer™ sensor, and it was verified that the sensor did not cause any discomfort when performing the different activities. One of the cameras was used to record participants’ facial expressions. Meanwhile, another camera was used to verify that the participants were doing the activities as well as to check that there were no problems during the experiment. At the end of the experiment, participants were separated in another room where they could not have contact with the participants who had not participated in the experiment yet; this prevented the spread of information that may cause a bias regarding the participants’ interactions.

### 5.4. Data Post-Processing

When the experiments were over, a post-processing of the acquired data was needed to filter non-relevant data as well as to synchronize the recordings of the interaction with the physiological signals. The first step was to clean up and remove data in which participants’ facial expressions were not captured due to the fact that during the activities, they accidentally covered their face, they were out of the field of view of the camera, or the angle of the face did not allow capturing all the facial expressions, leading to a potential misclassification of the emotions reflected in the face. Once the data were cleaned, a synchronization process was conducted to correlate facial expressions with the signals acquired by the Shimmer™ sensor. The synchronized data were processed by the iMotions™ software in order to determine the emotions that were presented by the participants during the experiment. The data related to the emotions were synchronized with the recordings of the interaction between the participants and the detergent bottles. This synchronization allowed finding key moments during the interaction. Finally, the environmental variables were included, although these variables were constant most of the time.

### 5.5. Data Interpretation and Analysis

Once the data were processed and synchronized, analysis and interpretation steps were required to generate a UX report. For this experiment, the following emotions were considered: joy, anger, surprise, and disgust. These emotions were selected due to the impact they have on customers’ behavior when buying a new product. The results presented in Table 2, Table 3 and Table 4 indicated the most prominent emotion during the interaction based on the analysis of facial expressions. Those results were contrasted by the self-reported answers provided by the participants at the end of each test.

The emotional response results of the participants that interacted with the bottles of Group A are presented in Table 2. As it can be observed, confusion can arise when comparing emotions such as anger and joy at the same time. However, when making a deeper analysis, the emotion recognition process could lead to identifying a concentration with anger, since they share a lot of the facial gestures. It can be observed that bottle 1 had the most positive impact in terms of utility, while bottle 2 has the greatest impact regarding aesthetic aspects.

The emotional response results of the participants that interacted with the bottles of Group B are presented in Table 3. Similar to the case of Group A, the emotion recognition system tended to confuse concentration with anger while interacting with the bottles. In this group, bottle 2 has the most positive impact regarding both aesthetic and practical use. Only bottle 1 had a better response in the case of first impressions.

The emotional response results of the participants that interacted with the bottles of Group C are presented in Table 4. In this case, bottle 1 had the most acceptance by the participants due to the joy presented when manipulating that bottle. Meanwhile, the disgust presented in bottle 2 could have been caused by its usability.

## 6. Discussion

KNX is an open-source communication protocol; this means that different types of applications can be developed that involve domotic systems. This provides the possibility of integrating KNX-based systems with systems not strictly related to domotic applications. This served as a starting point for the ED research line focusing on the use of smart systems beyond the original vision that society has about smart homes. As can be seen in the previous work [10,11,15] and other works of the same nature [12,13,14], home automation allows the modulation of environmental variables within a inhabitable space to generate the ideal conditions in which the inhabitants have a sense of comfort and well-being that, as a result, decreases their stress levels.

On the other hand, the generation of these conditions is related to the topic of UX. This opens the possibility of using KNX technology to recreate environments in which the user has greater immersion when interacting with a product or service. Although the original purpose of the testing booth presented in Section 3.1 was related to generating the ideal conditions for an individual based on the emotional response, the possibility of implementing it in experiments related to UX research was explored. Meanwhile, the use of emotion recognition tools used in this research allows monitoring the emotional behavior of the user during the interaction with the products and services that would be evaluated, obtaining a more genuine response about the expectations that the user has when interacting with the product or service; this leads to a more objective UX evaluation. With the framework presented in Section 4, a synchronization of the data obtained from the environmental sensors within the testing booth can be carried out with the biometric data of the users obtained during the interaction, facilitating the identification of key moments that influence the acceptance of a product or service through their emotional response.

The experiment presented in Section 5 served to validate that the framework of the testing booth can be used for UX-related experimentation. This can be verified, since the testing booth could be adapted to generate a scenario according to the context of the product through the control of KNX actuators, in this case a laundry room for the evaluation of the detergent bottle design. However, this framework could be replicated and adapted to other domotic technologies, but that could increase the complexity of the framework. On the other hand, the analysis of facial expressions to determine the user’s emotional response in conjunction with the recordings of the interaction was carried out with the commercial software iMotions™. The use of this software had the purpose of not having any type of uncertainty when analyzing the user’s emotional response, since it is a software with validated models. As future work, we plan to develop our own facial analysis model as well as use tools such as OpenPose [61] to monitor the user’s body language. By obtaining the emotional response of the user and carrying out the processing corresponding to the synchronization of the data, it can be interpreted which factor of the product influenced the emotional response of the user either for better or worse. All this information was collected to generate a report about the emotional behavior of the users.

This was a first approach to use the tools from the ED research line in experiments related to UX. The use of the testing booth can be part of the design of a methodology for the objective UX evaluation. However, the limitations of emotion recognition tools must be considered, since it is mainly based on the analysis of facial expressions. This means that if the activities to be carried out include those in which the face is totally or partially obstructed, the emotional response during the interaction with the product to be evaluated will not be able to be identified. Although wearables are also used to obtain GSR and heart rate, these signals serve more as a complement for the analysis of facial expressions; these signals alone cannot provide a prediction about the user’s emotional response. Finally, the ethical aspect must also be considered when collecting data related to the user’s emotional response, since they are sensitive data that can be used for other types of purposes.

## Figures and Tables

**Figure 1 sensors-23-09570-f001:**
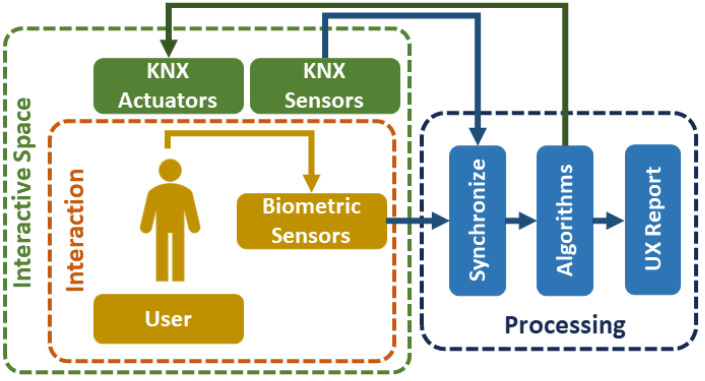
Diagram of the general solution.

**Figure 2 sensors-23-09570-f002:**

General process for facial expression analysis for emotion recognition based on computer vision systems [42].

**Figure 3 sensors-23-09570-f003:**
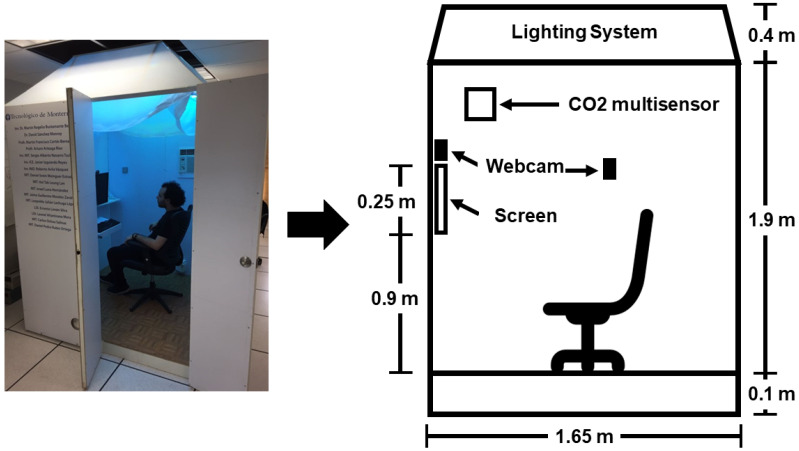
Emotional Domotics testing booth.

**Figure 4 sensors-23-09570-f004:**
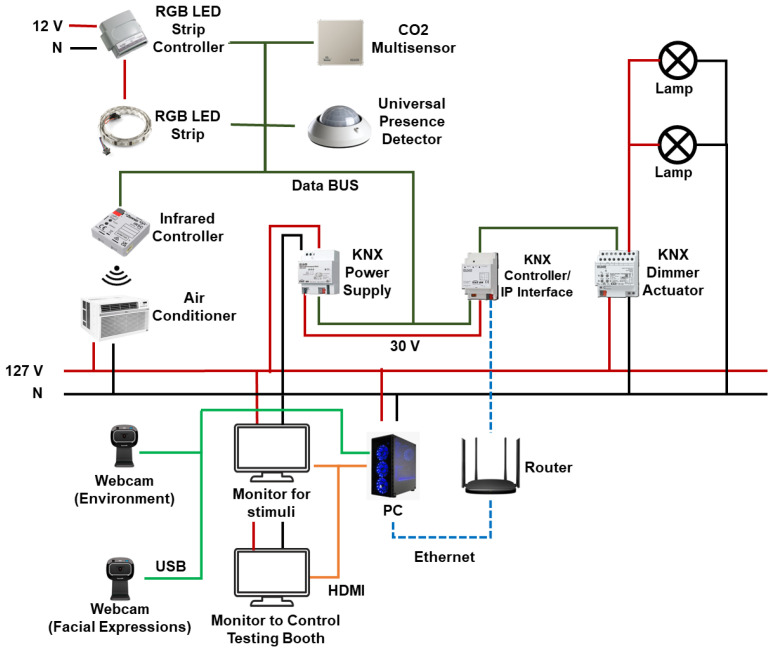
Testing booth wiring diagram.

**Figure 5 sensors-23-09570-f005:**
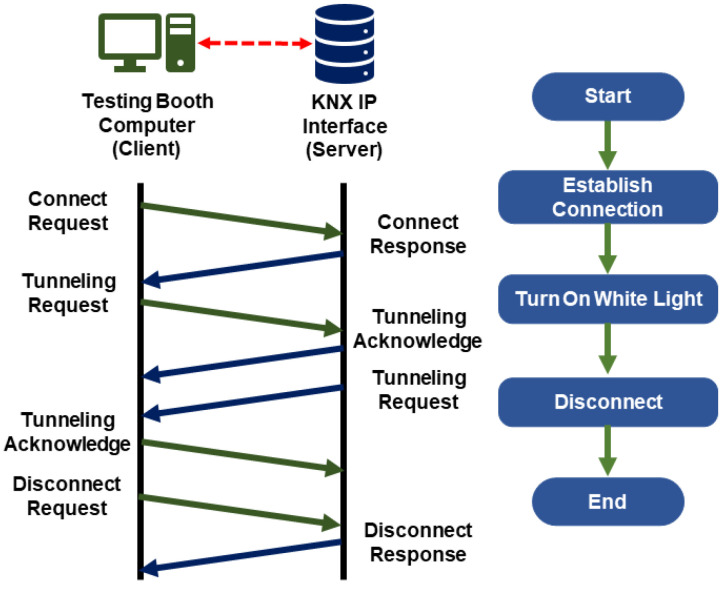
KNX telegrams transmission and reception to turn on a light.

**Figure 6 sensors-23-09570-f006:**
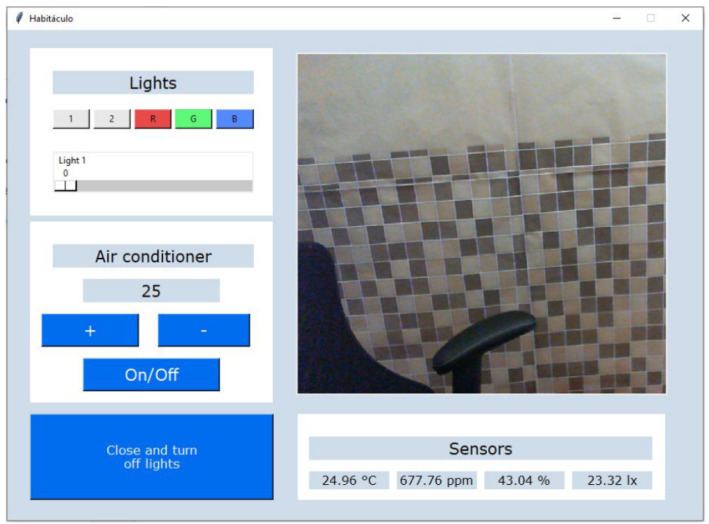
GUI to monitor the inside of the testing booth.

**Figure 7 sensors-23-09570-f007:**
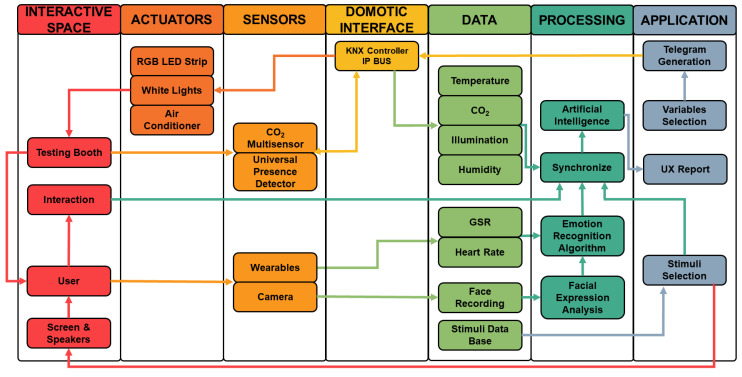
Framework for the communication system of the testing booth.

**Figure 8 sensors-23-09570-f008:**
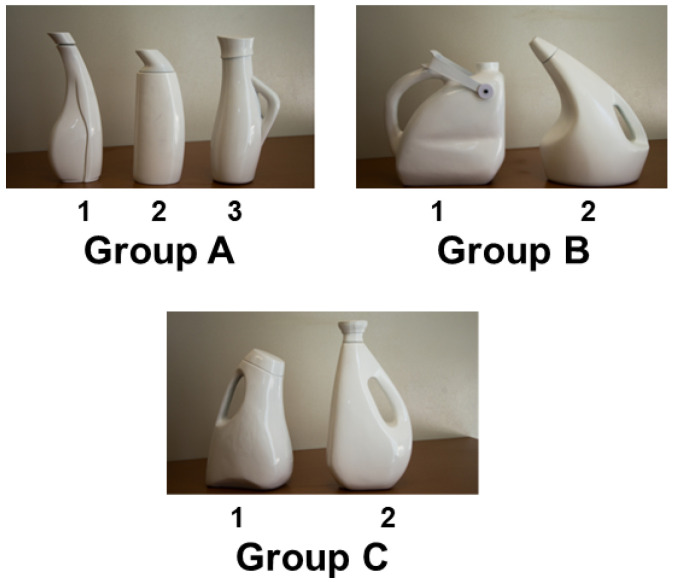
Group A: 3 bottles of 600 mL. Group B: 2 bottles of 5 L. Group C: 2 bottles of 1 gallon.

**Table 1 sensors-23-09570-t001:** Participants’ distribution in each detergent bottle group.

	Group A	Group B	Group C
# Female	10	10	7
# Male	0	0	3
Age	18–26 years	36–43 years	36–43 years

**Table 2 sensors-23-09570-t002:** Results from Group A.

	Joy	Anger	Surprise	Disgust
First impression	Bottle 3	Bottle 3	Bottle 1	Bottle 3
Dosage measurement	Bottle 1	Bottle 1	Bottle 1	Bottle 3
Bottle’s grip	Bottle 1	Bottle 1	Bottle 3	Bottle 3
Spill prevention	Bottle 3	Bottle 3	Bottle 3	Bottle 2
Residue prevention	Bottle 3	Bottle 3	Bottle 3	Bottle 3

**Table 3 sensors-23-09570-t003:** Results from Group B.

	Joy	Anger	Surprise	Disgust
First impression	Bottle 1	Bottle 1	Bottle 2	Bottle 2
Dosage measurement	Bottle 2	Bottle 2	Bottle 2	Bottle 2
Bottle’s grip	Bottle 2	Bottle 2	Bottle 2	Bottle 1
Spill prevention	Bottle 2	Bottle 2	Bottle 2	Bottle 1
Residue prevention	Bottle 2	Bottle 2	Bottle 1	Bottle 1

**Table 4 sensors-23-09570-t004:** Results from Group C.

	Joy	Anger	Surprise	Disgust
First impression	Bottle 1	Bottle 2	Bottle 2	Bottle 2
Dosage measurement	Bottle 2	Bottle 2	Bottle 2	Bottle 2
Bottle’s grip	Bottle 1	Bottle 2	Bottle 1	Bottle 1
Spill prevention	Bottle 1	Bottle 1	Bottle 1	Bottle 2
Residue prevention	Bottle 1	Bottle 2	Bottle 1	Bottle 2

## Data Availability

Data are contained within the article.

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
