# Peer review of "Development of a Framework for the Communication System Based on KNX for an Interactive Space for UX Evaluation"

_sensors, 2023, doi:10.3390/s23239570_

Round 1

Reviewer 1 Report

Comments and Suggestions for Authors

Dear Authors,

I found your paper “Development of a Framework for the Communication System Based on KNX for an Interactive Space for UX Evaluation” relevant to the fields of home automation and user experience (UX), presenting the model for the communication system of the testing booth that integrates KNX systems and emotion recognition systems. The paper is well written, with explicit background and results that are possibly important for future research, but the paper itself does not follow the Sensors journal template (https://www.mdpi.com/files/word-templates/sensors-template.dot ) and does not provide some key information that readers expect to find:

1. The Abstract should follow the following structure “(1) Place the question addressed in a broad context and highlight the purpose of the study; (2) briefly describe the main methods or treatments applied; (3) summarize the article’s main findings; (4) indicate the main conclusions or interpretations.” At this moment I see only a summary of what was done, but the purpose of the study is not clearly stated, nor are the main findings and conclusions provided. Did you manage to solve the problem? How good is your solution?

2. The Introduction lacks related works. Although you made a separate Background section, with a sub-section called “2.5. Previous Work in Emotional Domotics”, it only cites your own previous studies. Therefore, it is unclear what the results of other researchers are. Are you the first and only ones to carry out research activities in this area?

·        Please, add references with related works and evaluate their findings/results. What are the limitations of previous studies? Why do you need to do your own research? If there are no related works in this area, please state this explicitly in the text.

·        Clearly describe the problem you and other researchers are solving. Is it related to KNX restrictions (“KNX systems are not compatible with the emotion recognition tools”)?

·        Again, you may follow the journal template, which suggests the following introduction content: “The introduction should briefly place the study in a broad context and highlight why it is important. It should define the purpose of the work and its significance. The current state of the research field should be carefully reviewed and key publications cited. Please highlight controversial and diverging hypotheses when necessary. Finally, briefly mention the main aim of the work and highlight the principal conclusions.“

3. Some figure and table captions should be more specific and reflect the content of the figure or table. For example:

·        “Figure 7. Proposed framework.” – What kind of framework? What is the Framework for?

·        “Figure 8. 7. Group A: 600 mL. Group B: 1 gallon. Group C: 5 L.” So, is it 8 or 7? What does Group mean? What kind of group?

·        “Table 2. Results from group A.” – What kind of results? Emotional response?

and so on.

4. The article lacks the Discussion section, where “authors should discuss the results and how they can be interpreted from the perspective of previous studies and of the working hypotheses. The findings and their implications should be discussed in the broadest context possible. Future research directions may also be highlighted.“ The Conclusions section also does not contain this information and is not actually mandatory according to the Sensors template. Add the Discussions section, describe how your solution helps solve the problem, and compare your findings with the results of previous studies.

Comments on the Quality of English Language

The quality of the English language is good. I found only minor spelling errors and typos, which can be easily corrected with spellcheckers.

Author Response

Dear reviewer, thank you for your comments.

I made the following changes based on your comments:

  1. Abstract was changed according to the structure provided by the MDPI – Sensors journal.
  2. The introduction was modified in order to include relevant studies from other authors (these are not covered in the 2. Background section, just in the Introduction section). The changes were based on the journal template. The part about the compatibility between KNX and emotion recognition sensors was modified to explain that KNX was not designed to for emotion recognition sensors, but it is not impossible to include them using different tools such as Python.
  3. More context was added for Figures 7 and 8, and Tables 2, 3, and 4.
  4. The Conclusion section was replaced by the Discussion section. The Discussion section follows the structure according to the template. Main findings and limitations of our research were added.

I hope that the revised version fulfills your requirements.

Reviewer 2 Report

Comments and Suggestions for Authors

The Development of a Framework for the Communication System Based on KNX as discussed :

  1. The aims as title to develop framework but in the paper not much intention to discuss how the framework work and what are issues to current framework, no clear problem statement in Introduction or literature review.
  2. The framework as proposed should elaborate in detail and clearly discuss step by step start from human sensing system as shows in figure 3 Emotional Domotics testing booth. Statement the testing booth has a webcam to record the facial expressions of the participants in booth order to analyze them and identify emotions, the web camera and detail specification should mention and discuss.
  3. The use KNX in section 2.2 should express the characteristic of KNX technology and suitable apply to human emotion. How relationship to the human emotional analysis ? as in abstract mention KNX systems are not compatible with the emotion recognition tools used in this research, need justification and how much contribute to the framework, if so ?
  4. Figure 3 shows a testing booth should explain in detail for example how the size and where the camera installed and how many, the purpose and parameters to be capture, etc.
  5. Figure 4 shows the testing booth wiring diagram but no web camera as mention to capture human expression.
  6. Figure 1,3,5,6,8 and should be center of page.
  7. Figure 6 mention the time was set due to there are not abrupt changes in the environmental variables during a short period of time, assume to refer human expression on booth but have to detail discussed on every devices as shows in figure 6.
  8. Figure 8 what is relation to the KNX framework, how the detergent bootle related to human express, have to elaborate more detail.
  9. Conclusion have to realize to the results and what is main contribution of the designed framework use KNX technology.
  10. Presentation of the manuscript such as figure should be improve with high quality.

Author Response

Dear reviewer, thank you for your comments.

I made the following changes based on your comments:

  1. The introduction was modified in order to establish the problem statement and related works according to the Sensors template.
  2. The testing booth was detailed.
  3. Changes were made to explain that that KNX was not designed to for emotion recognition sensors, but it is not impossible to include them using different tools such as Python.
  4. Changes about the dimensions and instruments of the testing booth were added in Figure 3.
  5. Changes about the wiring diagram including the webcams and PC to control the testing booth were added in Figure 4.
  6. Figures were centered.
  7. It is mentioned the differences between the KNX system acquisition period and the frame rate of the webcam for emotion recognition.
  8. Section 5 describes the process about the framework presented in Figure 7.
  9. The Conclusion section was changed to be a Discussion section. In this section, it is explained the main findings and limitations of the work presented in the manuscript. All these changes were made according to the Sensors template.
  10. Changes were made according to the Sensors template. Some figures were enhanced to have a better resolution.

I hope that the revised version fulfills your requirements.

Round 2

Reviewer 1 Report

Comments and Suggestions for Authors

Dear Authors,

Thank you for addressing all comments and suggestions, which were provided. Interesting research was conducted and important results were achieved.

Congratulations!

Comments on the Quality of English Language

English language quality is good. I see only few grammatical errors, which can be easily corrected.